# Learning Continuous-Time Dynamics by Stochastic Differential Networks

## Abstract

Learning continuous-time stochastic dynamics is a fundamental and essential problem in modeling sporadic time series, whose observations are irregular and sparse in both time and dimension. For a given system whose latent states and observed data are multivariate, it is generally impossible to derive a precise continuous-time stochastic process to describe the system behaviors. To solve the above problem, we apply Variational Bayesian method and propose a flexible continuous-time stochastic recurrent neural network named *Variational Stochastic Differential Networks (VSDN)*, which embeds the complicated dynamics of the sporadic time series by neural Stochastic Differential Equations (SDE). VSDNs capture the stochastic dependency among latent states and observations by deep neural networks. We also incorporate two differential Evidence Lower Bounds to efficiently train the models. Through comprehensive experiments, we show that VSDNs outperform state-of-the-art continuous-time deep learning models and achieve remarkable performance on prediction and interpolation tasks for sporadic time series.

## 1 Introduction and Related Works

Many real-world systems experience complicated stochastic dynamics over a continuous time period. The challenges on modeling the stochastic dynamics mainly come from two sources. First, the underlying state transitions of many systems are often uncertain, as they are placed in unpredictable environment with their states continuously affected by unknown disturbances. Second, the monitoring data collected may be sparse and at irregular intervals as a result of the sampling strategy or data corruption. The sporadic data sequence loses a large amount of information and system behaviors hidden behind the intervals of the observed data. In order to accurately model and analyze dynamics of these systems, it is important to reliably and efficiently represent the continuous-time stochastic process based on the discrete-time observations.

In some domains, the derivation of the continuous-time stochastic model relies heavily on human knowledge and many studies focus on its inference problem (Ryder et al., 2018; Särkkä et al., 2015). But in more domains (e.g., video analysis (Vondrick et al., 2016) and human activity detection (Rubanova et al., 2019)), it is difficult and sometimes intractable to derive an accurate model to capture the underlying temporal evolution from the collected sequence of data. Although some studies have been made on approximating the stochastic process from the data collected, the majority of these methods define the system dynamics with a linear model (Macke et al., 2011; Yu et al., 2009b;a), which can not well represent high-dimensional data with nonlinear relationship. Recently, the Neural Ordinary Differential Equation (ODE) studies (Chen et al., 2018; Rubanova et al., 2019; Jia & Benson, 2019; De Brouwer et al., 2019; Yildiz et al., 2019; Kidger et al., 2020) introduce deep learning models to learn an ODE and apply it to approximate continuous-time dynamics. Nevertheless, these methods generally neglect the randomness of the latent state trajectories and posit simplified assumptions on the data distribution (e.g. Gaussian), which strongly limits their capability of modeling complicated continuous-time stochastic processes.

Compared to ODE, Stochastic Differential Equation (SDE) (Jørgensen et al., 2020) is a more practical solution in modeling the continuous-time stochastic process. Recently there have been some studies on bridging the gap between deep neural networks and SDEs (Ha et al., 2018). In (Hegde et al., 2019; Liu et al., 2020; Peluchetti & Favaro, 2020; Wang et al., 2019; Kong et al., 2020), SDEs

are introduced to define more robust and accurate deep learning architectures for supervised learning problems (e.g. classification and regression). These studies focus on the design of neural network architectures, and are orthogonal to our work on the modeling of sporadic time series. In (Tzen & Raginsky, 2019a;b) the authors studied the theoretical guarantees of the optimization and inference problems of Neural SDEs. In (Li et al., 2020), a stochastic adjoint method is proposed to efficiently compute the gradients for neural SDEs.

In this paper, we propose a new continuous-time stochastic recurrent network called **Variational Stochastic Differential Network (VSDN)** that incorporates SDEs into recurrent neural model to effectively model the continuous-time stochastic dynamics based only on sparse or irregular observations. Taking advantage of the capacity of deep neural networks, VSDN has higher flexibility and generalizability in modeling the nonlinear stochastic dependency from high-dimensional observations.

Compared to Neural ODEs, VSDN incorporates the latent state trajectory to capture the underlying factors of the system dynamics. The trajectory helps to more flexibly model the data distribution and more accurately generate the output data than Neural ODEs. Parallel to the theoretical analysis (Tzen & Raginsky, 2019a;b) and gradient computations (Li et al., 2020), our study focuses more on exploring the feasible variational loss and flexible recurrent architecture for the Neural SDEs to model the sporadic data.

The contributions of this paper are three-fold:

1. We incorporate the continuous-time variants of VAE and IWAE losses into VSDN to train the continuous-time stochastic neural networks with latent state trajectories.

2. We propose the efficient and flexible network architecture of VSDN which can model the complicated stochastic process under high-dimensional sporadic data sequences.

3. We conduct comprehensive experiments to show that VSDN outperforms state-of-the-art deep learning methods on modeling the continuous-time dynamics and achieves remarkable performance in the prediction and interpolation of irregular or sporadic time series.

The rest of this paper is organized as follows. In Section 2, we first present the continuous-time variants of VAE loss, and then derive a continuous-time IWAE loss to train continuous-time state-space models with deep neural networks. In Section 3, we propose the deep learning structures of VSDN. Comprehensive experiments are presented in section 4 and conclusion is given in section 5.

## 2 CONTINUOUS-TIME VARIATIONAL BAYES

In this section, we first introduce the basic notations and formulate our problem. We then define the continuous-time variants of the Variational Auto-Encoding (VAE) and Importance-Weighted Auto-Encoding (IWAE) lower bounds to enable the efficient training of our models. Due to the page limit, we present all deductions in Appendix A.

### 2.1 BASIC NOTATIONS AND PROBLEM FORMULATION

Throughout this paper, we define $X_t \in \mathbb{R}^{d_1}$ as the continuous-time latent state at time $t$ and $Y_n \in \mathbb{R}^{d_2}$ as the $n_{th}$ discrete-time observed data at time $t_n$. $d_1$ and $d_2$ are the dimensions of the latent state and observation, respectively. $X_{<t}$ is the continuous trajectory before time $t$ and $X_{\leq t}$ is the path up to time $t$. $Y_{n_1:n_2}$ is the sequence of data points and $X_{t_{n_1}:t_{n_2}}$ is the continuous-time state trajectory from $t_{n_1}$ to $t_{n_2}$. $\mathcal{Y}_t = \{Y_n | t_n < t\}$ is the historical observations before $t$ and $\mathbb{Y}_t = \{Y_n | t_n \geq t\}$ is the current and future observations. For simplicity, we also assume that the initial value of the latent state is constant. The results in this paper can be easily extended to the situation that the initial states are also random variables. Given $K$ data sequences $\{y_{1:n_i}^{(i)}\}, i = 1, \cdots, K$, the target of our study is to learn an accurate continuous-time generative model $\mathcal{G}$ that maximizes the log-likelihood:

$$\mathcal{G} = \arg\max_{\mathcal{G}} \frac{1}{K} \sum_{i=1}^{K} \log P_{\mathcal{G}}(y_{1:n_i}^{(i)}). \tag{1}$$

For Multivariate sequential data, there exists a complicated nonlinear relationship between the observed data and the unobservable latent state, which can be either the physical state of a dynamic system or the low-dimensional manifold of data. In our study, the latent state evolves in the continuous time domain and generates the observation through some transformation.

## 2.2 Continuous-Time Variational Inference

In order to capture the underlying stochastic process from sporadic data, we design the generative model as a neural continuous-time state-space model, which consists of a latent Stochastic Differential Equation (SDE) and a conditional distribution of the observation. The latent SDE describes the stochastic process of the latent states and the conditional distribution depicts the probabilistic dependency of the current data with the latent states and historical observations:

$$dX_t = H_{\mathcal{G}}(X_t, \mathcal{Y}_t; t)dt + R_{\mathcal{G}}(\mathcal{Y}_t; t)dW_t, \tag{2}$$

$$P_{\mathcal{G}}(Y_n | Y_{1:n-1}, X_{t_n}) = \Phi(Y_n | f(Y_{1:n-1}, X_{t_n})), \tag{3}$$

where $H_{\mathcal{G}}$ and $R_{\mathcal{G}}$ are the drift and diffusion functions of the latent SDE. $W_t$ denotes the a Wiener process, which is also called standard Brownian motion. To integrate the information of the observed data, $H_{\mathcal{G}}$ is the function of the current state $X_t$ and the historical observations $\mathcal{Y}_t$. However, $R_{\mathcal{G}}$ only uses the historical data as input. It is not beneficial to include $X_t$ as the input of the diffusion function, as it will inject more noise into gradients of the network parameters. A detailed example and analysis of the noise injection problem is given in Appendix B. $\Phi(\cdot)$ is a parametric family of distributions over the data and $f(\cdot)$ is the function to compute the parameters of $\Phi$. With the advance of deep learning methods, we parameterize $H_{\mathcal{G}}$, $R_{\mathcal{G}}$ and $f(\cdot)$ by deep neural networks.

**Continuous-Time Auto-Encoding Variational Bayes:** The exact log-likelihood of the generative model is given as

$$\log P_{\mathcal{G}}(y_{1:n}) = \log \int P_{\mathcal{G}}(X_{\leq t_n}) \prod_{i=1}^{n} P_{\mathcal{G}}(y_i | y_{1:n-1}, X_{t_i}) dX_{\leq t_n} \tag{4}$$

which does not have the closed-form solution in general. Therefore, $\mathcal{G}$ can not be directly trained by maximizing log-likelihood. To overcome this difficulty, an inference model $\mathcal{Q}$ is introduced to depict the stochastic dependency of the latent state on observed data. Similar to the generative model, $\mathcal{Q}$ consists of a posterior SDE:

$$dX_t = H_{\mathcal{Q}}(X_t, \mathcal{Y}_t, \mathbb{Y}_t; t)dt + R_{\mathcal{G}}(\mathcal{Y}_t; t)dW_t, \tag{5}$$

where $H_{\mathcal{Q}}$ is the posterior drift function. Different from $H_{\mathcal{G}}$, $H_{\mathcal{Q}}$ also uses the future observation $\mathbb{Y}_t$ as the input and therefore the inference model $\mathcal{Q}$ induces the posterior distribution $P_{\mathcal{Q}}(X_{\leq t_n} | y_{1:n})$.

Based on Auto-Encoding Variational Bayes (Kingma & Welling, 2014), it is straightforward to introduce a continuous-time variant of the VAE lower bound of the log-likelihood:

$$\mathcal{L}_{VAE}(y_{1:n}) = -\beta KL(P_{\mathcal{Q}} || P_{\mathcal{G}}) + \sum_{i=1}^{n} \mathbb{E}_{P_{\mathcal{Q}}(X_{t_i})} \log P_{\mathcal{G}}(y_i | y_{1:n-1}, X_{t_i}), \tag{6}$$

$$KL(P_{\mathcal{Q}} || P_{\mathcal{G}}) = \frac{1}{2} \int_0^{t_n} \mathbb{E}_{P_{\mathcal{Q}}(X_t)} \Big( (H_{\mathcal{Q}} - H_{\mathcal{G}})^T [R_{\mathcal{G}} R_{\mathcal{G}}^T]^{-1} (H_{\mathcal{Q}} - H_{\mathcal{G}}) \Big) dt. \tag{7}$$

where $P_{\mathcal{G}}(X_{\leq t_n})$ and $P_{\mathcal{Q}}(X_{\leq t_n})$ are the probability density of the latent states induced by the prior SDE Eq. (2) and the posterior SDE Eq. (5). $KL(\cdot || \cdot)$ denotes the KL divergence between two distributions and $\beta$ is a hyper-parameter to weight the effect of the KL terms. In this paper, we fix $\beta$ as 1.0 and $\mathcal{L}_{VAE}$ is the original VAE objective (Kingma & Welling, 2014). In $\beta$-VAE (Higgins et al., 2017; Burgess et al., 2018), it is shown that a larger $\beta$ can encourage the model to learn more efficient and disentangled representation from the data. Eq. (5) is restricted to having the same diffusion function as Eq. (2). A feasible $\mathcal{L}_{VAE}$ can not be defined to train VSDN-SDE without this restriction, as the KL divergence of two SDEs with different diffusions will be infinite (Archambeau et al., 2008).

The VAE objective has been widely used for discrete-time stochastic recurrent modals, such as LFADS (Sussillo et al., 2016), VRNN (Chung et al., 2015) and SRNN (Fraccaro et al., 2016). The

major difference between these models and our work is that we incorporate a continuous-time latent state into our model while the latent states of the discrete-time models evolve only at distinct and separate time slots.

**Continuous-Time Importance Weighted Variational Bayes:** $\mathcal{L}_{VAE}(y_{1:n})$ equals the exact log-likelihood when $P_{\mathcal{Q}}(X_{\leq t_n})$ of the inference model is identical to the exact posterior distribution induced by the generative model. The errors of the inference model can result in the looseness of the VAE loss for the model training. Under the framework of Importance-Weighted Auto-Encoder (IWAE) (Burda et al., 2016; Cremer et al., 2017), we can define a tighter evidence lower bound:

$$\widetilde{\mathcal{L}}_{IWAE}^K(y_{1:n}) = \mathbb{E}_{x_{\leq t_n}^1, \cdots, x_{\leq t_n}^K \sim P_{\mathcal{Q}}(x_{\leq t_n})} \Big( \log \frac{1}{K} \sum_{k=1}^{K} w_k \prod_{i=1}^{n} P_{\mathcal{G}}(y_i | y_{1:n-1}, X_{t_i}) \Big), \tag{8}$$

where the importance weights satisfy the following SDE:

$$d \log w_k = d \log \frac{P_{\mathcal{G}}(x_{\leq t_n}^k)}{P_{\mathcal{Q}}(x_{\leq t_n}^k)} = -\frac{1}{2}(H_{\mathcal{Q}} - H_{\mathcal{G}})^T [R_{\mathcal{G}} R_{\mathcal{G}}^T]^{-1}(H_{\mathcal{Q}} - H_{\mathcal{G}}) dt$$
$$- (H_{\mathcal{Q}} - H_{\mathcal{G}})^T [R_{\mathcal{G}}]^{-1} dW_t. \tag{9}$$

Given the variational auto-encoding lower bound $\mathcal{L}_{VAE}(\cdot)$ and the importance weighted auto-encoding lower bound $\mathcal{L}_{IWAE}^K(\cdot)$ for the continuous-time generative model, the tightness of the lower bounds are given by the following inequality:

$$\log P_{\mathcal{G}}(y_{1:n}) \geq \widetilde{\mathcal{L}}_{IWAE}^{K+1}(\cdot) \geq \widetilde{\mathcal{L}}_{IWAE}^K(\cdot) \geq \mathcal{L}_{VAE}(\cdot), \tag{10}$$

for any positive integer $K$. Consequently, $\widetilde{\mathcal{L}}_{IWAE}^K(\cdot)$ is infinite if the diffusions of Eq. (2) and Eq. (5) are different. In our implementation, we notice that the training of our models by $\widetilde{\mathcal{L}}_{IWAE}^K$ is not stable, possibly due to the drawbacks of importance sampling and the Signal-To-Noise problem (Rainforth et al., 2018). To alleviate the problem, we train our model by a convex combination of the VAE and IWAE losses:

$$\mathcal{L}_{IWAE}^K(y_{1:n}) = (1 - \alpha)\mathcal{L}_{VAE}(y_{1:n}) + \alpha \widetilde{\mathcal{L}}_{IWAE}^K(y_{1:n}), \quad \alpha \in (0, 1). \tag{11}$$

With the use of reparameterization (Kingma & Welling, 2014), both $\mathcal{L}_{VAE}(y_{1:n})$ and $\mathcal{L}_{IWAE}^K(y_{1:n})$ are differentiable with respect to the parameters of the generative and inference models. Therefore, they can be applied to train continuous-time stochastic models with deep learning components.

## 3 VARIATIONAL STOCHASTIC DIFFERENTIAL NETWORKS

We propose a new continuous-time stochastic recurrent network called *Variational Stochastic Differential Network (VSDN)* (Figure 1). VSDN introduces the latent state to capture the underlying unobservable factors that generate the observed data, and incorporates efficient deep learning structures to compute the components in the generative model Eq. (2) - (3) and inference model Eq. (5).

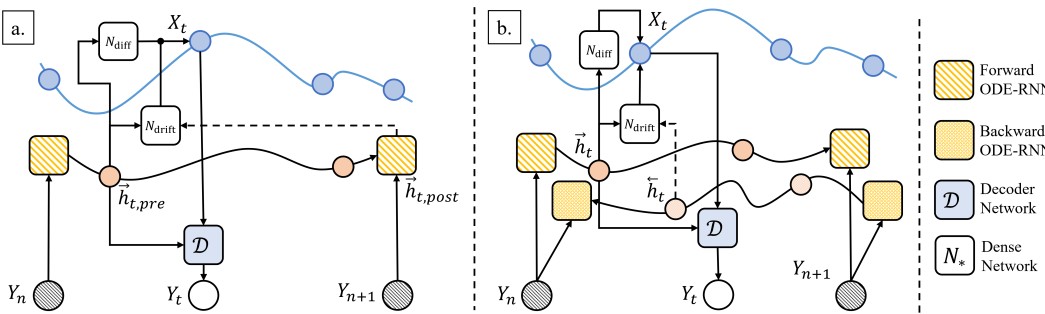

Figure 1: Model Architectures of (a) VSDN-F (filtering); (b) VSDN-S (smoothing).

**Generative Model $\mathcal{G}$:** Inside the generative model, the latent SDE Eq. (2) depicts the dynamics of the latent state trajectory controlled by the historical observations $\mathcal{Y}_t$. Both the drift and diffusion functions have the dependency on $\mathcal{Y}_t$. Therefore, we first apply a forward ODE-RNN (Rubanova et al., 2019) to embed the information of historical data into the hidden feature $\overrightarrow{h}_{t,pre}$. Two feed-forward networks are defined to compute drift and diffusion respectively. The decoder network further computes the parameters of the conditional distribution in Eq. (3) by the concatenation of the latent state and forward feature

$$H_{\mathcal{G}} = N_{drift}([X_t, \overrightarrow{h}_{t,pre} = \text{ODE-RNN}_1(\mathcal{Y}_t; t)]), \quad R_{\mathcal{G}} = \exp(N_{diff}(\overrightarrow{h}_{t,pre})),$$

$$P_{\mathcal{G}}(Y_t|\mathcal{Y}_t, X_t) = \Phi(Y_n|f = \mathcal{D}([X_t, \overrightarrow{h}_{t,pre}])). \tag{12}$$

**Inference Model $\mathcal{Q}$:** We propose two types of inference models in VSDN: a filtering model, and a smoothing model. $\mathcal{L}_{VAE}(y_{1:n})$ and $\mathcal{L}_{IWAE}^K(y_{1:n})$ equal the exact log-likelihood when $P_{\mathcal{Q}}(X_{\leq t_n})$ is identical to the exact posterior distribution $P_{\mathcal{G}}(X_{\leq t_n}|y_{1:n})$. The inference model must process the the whole data sequence to compute $H_{\mathcal{Q}}$ at a time. According to d-separation (Bishop, 2006), the latent state $X_t$ is dependent on both the historical data $\mathcal{Y}_t$ and future observations $\mathbb{Y}_t$. Therefore, we first define $\mathcal{Q}$ as a smoothing model by introducing a backward ODE-RNN to embed the information of the future observations into a hidden feature $\overleftarrow{h}_t$. The drift function is computed as:

$$H_{\mathcal{Q}} = N_{drift}([X_t, \overrightarrow{h}_{t,pre} + \overleftarrow{h}_t]), \quad \overleftarrow{h}_t = \text{ODE-RNN}_2(\mathbb{Y}_t; t)]). \tag{13}$$

In real-world applications, it is sometimes possible to have close performance in inference without processing the future observations. Besides, the future measurements are intractable in online systems. Therefore, we also design a filtering inference model that infers the latent state from the historical and current data. The drift of the filtering model is given as:

$$H_{\mathcal{Q}} = \begin{cases} N_{drift}([X_t, \overrightarrow{h}_{t,pre} + \overrightarrow{h}_{t,post}]) & \text{if there is } y_t \text{ at t} \\ H_{\mathcal{G}} & \text{otherwise} \end{cases} \tag{14}$$

where $\overrightarrow{h}_{t,post}$ is the post-observation updated feature of the forward ODE-RNN (Rubanova et al., 2019). The filtering model does not have to include a backward RNN to process the future observations and thus its running speed is faster.

The whole architectures of VSDN with filtering $\mathcal{Q}$ (VSDN-F) and smoothing $\mathcal{Q}$ (VSDN-S) are shown in Figure 1 (a) - (b). The inference model and the generative model share the drift network. This strategy can force the ODE-RNNs to embed more information into the hidden features and reduce the model complexity.

**Applications:** VSDN consists of a generative model and an inference model. The generative model is an online predictive model which can recurrently predict the future values of the sequence. The inference models can be applied to either filtering or smoothing problems of the latent states accordingly. Furthermore, the smoothing inference model infers the latent state trajectory from the whole sequence, which can be further used in Eq. (3) to synthesize missing data. Therefore, the smoothing inference model is capable of offline interpolation. The motivation of this paper is to design an efficient continuous-time stochastic recurrent model. Therefore, VSDNs only use the generative model to recurrently predict the future values in the experiments.

**Discussions:** VSDN has higher flexibility and model capability than current continuous-time deep learning models in modeling the sporadic sequences. LatentODE (Chen et al., 2018) and ODE$^2$VAE (Yildiz et al., 2019) encode the information of the time series into the initial values of the latent state trajectories and neglect the variance in the latent state transition. This strategy is impractical and inefficient in real-world applications, as it requires the initial latent states to disentangle the property of the long sequence. Furthermore, LatentODE, ODE$^2$VAE are offline models, as the encoder used during training of these models can not be directly used for online prediction. In contrast, VSDN defines a latent SDE controlled by the historical observations and recurrently integrates the information of the sequence along the time axis. It is more efficient than the initial state embedding and is also applicable in online prediction. GRU-ODE (De Brouwer et al., 2019), ODE-RNN (Rubanova et al., 2019) and NCDE (Kidger et al., 2020) also utilize recurrent scheme but does not explicitly model the stochasticity of the underlying latent state. Therefore, they are less capable than VSDN in modeling the complicated stochastic process of the irregular data.

# 4 EXPERIMENTS

In this section, we conduct comprehensive experiments to validate the performance of our models and demonstrate its advantages in real-world applications. We compare the performance of VSDN with state-of-the-art continuous-time recurrent neural networks (i.e. ODE-RNN (Rubanova et al., 2019) and GRU-ODE (De Brouwer et al., 2019)), LatentODE (Chen et al., 2018) and LatentSDE (Li et al., 2020).

## 4.1 HUMAN MOTION ACTIVITIES

We first evaluate the performance of different models on the prediction and interpolation problems for human motion capturing. For a given sequence of data points sampled at irregular time intervals, the prediction task is defined to estimate the next observed data in the time axis, and the interpolation task is defined to recover the missing parts of the whole data trajectory. In both prediction and interpolation tasks, only the generative models of VSDNs are evaluated. The experiments are conducted on the following datasets:

- **Human3.6M (Ionescu et al., 2014):** We apply the same data pre-processing as (Martinez et al., 2017), after which the data frame at each time is a 51-dimensional vector. The long data sequences are further segmented by 248 frames.

- **CMU MoCap**[*]**:** We follow the data pre-processing in (Liu et al., 2019). In each data frame, human activity is represented as a 62-dimensional vector and each dimension of the frames is normalized by global mean and standard deviation. The long data sequences are further segmented by 300 frames.

After data pre-processing, we randomly remove half of the frames in the data sequence as missing data. To quantify the model performance, we consider two evaluation metrics: one is the negative log-likelihood (NLL) per frame; the other is the frame-level mean square error (MSE) between the ground-true and estimated values. The model configurations are given in Appendix C. The model performance is shown in Tables 1 and 2.

VSDN incorporates SDE to model the stochastic dynamics, and also applies a recurrent structure to embed the information of the irregular time series into the whole latent state trajectory. With these advances, VSDN outperforms the baseline models in both the prediction and interpolation tasks. VSDN has much smaller negative log-likelihood, which indicates that it can better model the underlying stochastic process of the data. Furthermore, VSDN trained by IWAE losses has similar and sometimes better performance than those with VAE losses. As the latent state in the inference model has stochastic dependency on the future observations, VSDN-S using the smoothing model has slightly lower NLL and is a better choice than VSDN-F using filtering model.

Table 1: Model performance on Human3.6M dataset

|  | Prediction | | Interpolation | |
|---|---|---|---|---|
|  | NLL | MSE | NLL | MSE |
| LatentODE | $-45.34 \pm 0.85$ | $1.6421 \pm 0.041$ | $-45.31 \pm 0.85$ | $1.6477 \pm 0.041$ |
| LatentSDE | $-63.01 \pm 1.26$ | $0.8278 \pm 0.059$ | $-62.93 \pm 1.28$ | $0.8311 \pm 0.060$ |
| GRU-ODE | $-93.70 \pm 2.34$ | $0.3201 \pm 0.062$ | $-93.71 \pm 2.34$ | $0.3207 \pm 0.062$ |
| ODE-RNN | $-93.78 \pm 1.48$ | $0.2981 \pm 0.075$ | $-93.78 \pm 1.48$ | $0.2984 \pm 0.076$ |
| **VSDN-F (VAE)** | $-122.64 \pm 2.79$ | $0.2373 \pm 0.064$ | $-122.62 \pm 2.79$ | $0.2367 \pm 0.064$ |
| **VSDN-S (VAE)** | $-126.93 \pm 3.35$ | $0.2374 \pm 0.086$ | $-126.88 \pm 3.35$ | $0.2368 \pm 0.086$ |
| **VSDN-F (IWAE)** | $-125.55 \pm 6.64$ | $\mathbf{0.1751 \pm 0.092}$ | $-125.51 \pm 6.62$ | $\mathbf{0.1746 \pm 0.092}$ |
| **VSDN-S (IWAE)** | $\mathbf{-127.12 \pm 5.19}$ | $0.1797 \pm 0.073$ | $\mathbf{-127.08 \pm 5.17}$ | $0.1790 \pm 0.073$ |

---

[*]http://mocap.cs.cmu.edu/

Table 2: Model performance on MoCap dataset

| | Prediction | | Interpolation | |
|---|---|---|---|---|
| | NLL | MSE | NLL | MSE |
| LatentODE | $14.99 \pm 1.64$ | $49.51 \pm 0.38$ | $14.91 \pm 1.81$ | $49.88 \pm 0.39$ |
| LatentSDE | $-59.83 \pm 2.13$ | $30.11 \pm 0.28$ | $-60.13 \pm 2.59$ | $30.43 \pm 0.31$ |
| GRU-ODE | $-51.83 \pm 0.48$ | $32.77 \pm 0.11$ | $-51.91 \pm 0.49$ | $33.17 \pm 0.12$ |
| ODE-RNN | $-51.75 \pm 1.16$ | $31.81 \pm 0.21$ | $-51.82 \pm 1.18$ | $33.22 \pm 0.21$ |
| **VSDN-F (VAE)** | $-110.71 \pm 3.92$ | $20.64 \pm 0.39$ | $-111.40 \pm 3.88$ | $20.95 \pm 0.39$ |
| **VSDN-S (VAE)** | $-114.31 \pm 4.44$ | $\mathbf{19.05 \pm 0.59}$ | $-114.97 \pm 4.40$ | $\mathbf{19.35 \pm 0.59}$ |
| **VSDN-F (IWAE)** | $-109.84 \pm 5.32$ | $20.47 \pm 0.53$ | $-110.54 \pm 5.33$ | $20.76 \pm 0.53$ |
| **VSDN-S (IWAE)** | $\mathbf{-114.57 \pm 2.58}$ | $19.84 \pm 0.18$ | $\mathbf{-115.24 \pm 2.50}$ | $20.12 \pm 0.18$ |

**Visualization:** We further compare different models qualitatively through the visualization of the interpolated human skeletons in Figure 2. VSDN models are able to generate vivid skeletons that are closer to the ground-true ones. Instead, ODE-RNN and GRU-ODE can not interpolate the postures correctly (e.g the angles of arms in each frame are significantly different from the ground-true ones). We also observe that the motions generated by VSDNs are smooth and closer to the real data, while there are a large vibration in the movements generated by the baseline models. The videos of these human motions are provided in supplementary materials.

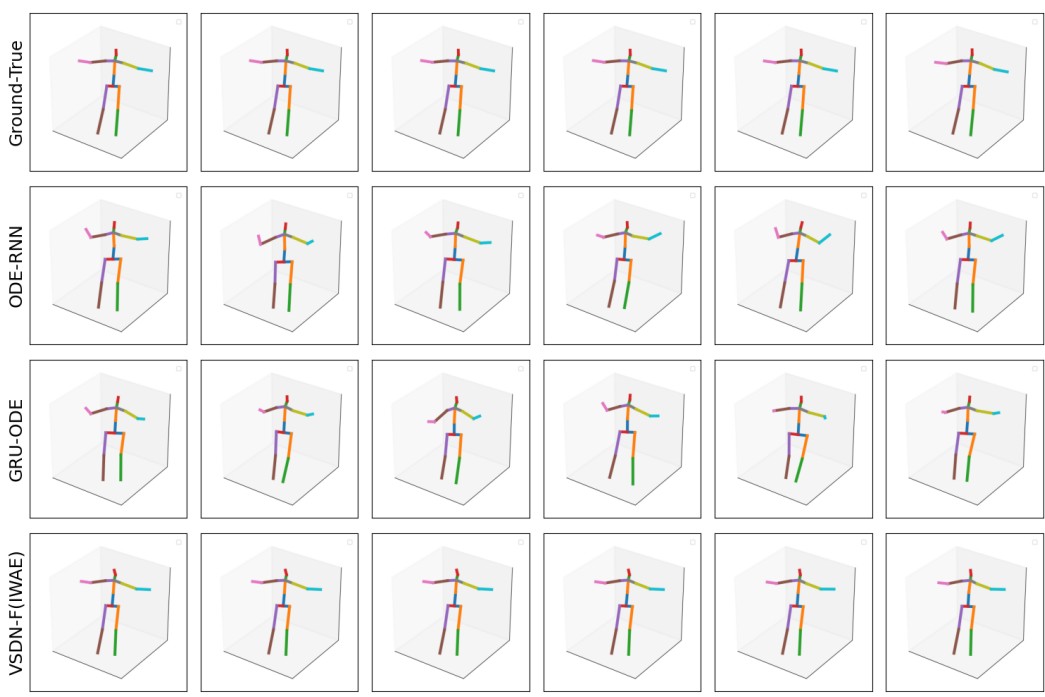

Figure 2: Visualization for human skeleton interpolation of different models.

## 4.2 TOY SIMULATION AND CLIMATE PREDICTION

We conduct additional experiments on two sporadic time series datasets in (De Brouwer et al., 2019):

- Double-OU[†]: The Double-OU dataset consists of data sequences synthesized by a 2-dimensional Ornstein-Uhlenbeck process, which is a classic stochastic differential equations in finance and physics.

---

[†]https://github.com/edebrouwer/gru_ode_bayes

- USHCN[‡]: The United State Historical Climatology Network (USHCN) dataset contains daily measurements of 5 climate variables from the meteorological stations in United States. In our experiment, we use the pre-processed subset of the data given in (De Brouwer et al., 2019).

Compared with the previous experiments, the data in Double-OU and USHCN are not only sampled at irregular times, but also have missing dimensions at each sampled frames. That is a data sequence is sparse in both time axis and frame dimension. We evaluate the model performance in predicting future values based on the sporadic observations.

The results are shown in Table 3. All VSDN models outperform the baseline ones. On the USHCN dataset, VSDN-S has better NLL than VSDN-F when using either VAE or IWAE losses in the training processes. VSDNs trained by IWAE loss also have smaller NLL than those trained by VAE loss. However, when running on the Double-OU dataset, the training with IWAE performs slightly worse than the training using the VAE loss. This is possibly caused by the randomness of the training process, as Double-OU process is a very simple stochastic differential equation and all VSDNs have the smallest errors in the prediction tasks.

Table 3: Model Performance on Sporadic Time Series

|  | Double-OU | | USHCN | |
|---|---|---|---|---|
|  | NLL | MSE | NLL | MSE |
| LatentODE | $0.351 \pm 0.023$ | $0.1201 \pm 0.0071$ | $1.319 \pm 0.156$ | $0.772 \pm 0.099$ |
| LatentSDE | $0.334 \pm 0.010$ | $0.1118 \pm 0.0029$ | $1.304 \pm 0.083$ | $0.748 \pm 0.116$ |
| GRU-ODE | $-0.997 \pm 0.021$ | $0.0080 \pm 0.0004$ | $0.940 \pm 0.058$ | $0.443 \pm 0.067$ |
| ODE-RNN | $-1.002 \pm 0.014$ | $0.0082 \pm 0.0003$ | $0.866 \pm 0.057$ | $0.397 \pm 0.064$ |
| **VSDN-F (VAE)** | $\mathbf{-1.145 \pm 0.029}$ | $\mathbf{0.0065 \pm 0.0003}$ | $0.736 \pm 0.111$ | $0.384 \pm 0.060$ |
| **VSDN-S (VAE)** | $\mathbf{-1.145 \pm 0.013}$ | $\mathbf{0.0065 \pm 0.0002}$ | $0.716 \pm 0.113$ | $0.390 \pm 0.057$ |
| **VSDN-F (IWAE)** | $-1.143 \pm 0.027$ | $\mathbf{0.0065 \pm 0.0004}$ | $0.661 \pm 0.096$ | $\mathbf{0.370 \pm 0.062}$ |
| **VSDN-S (IWAE)** | $-1.139 \pm 0.019$ | $\mathbf{0.0065 \pm 0.0003}$ | $\mathbf{0.654 \pm 0.084}$ | $0.381 \pm 0.058$ |

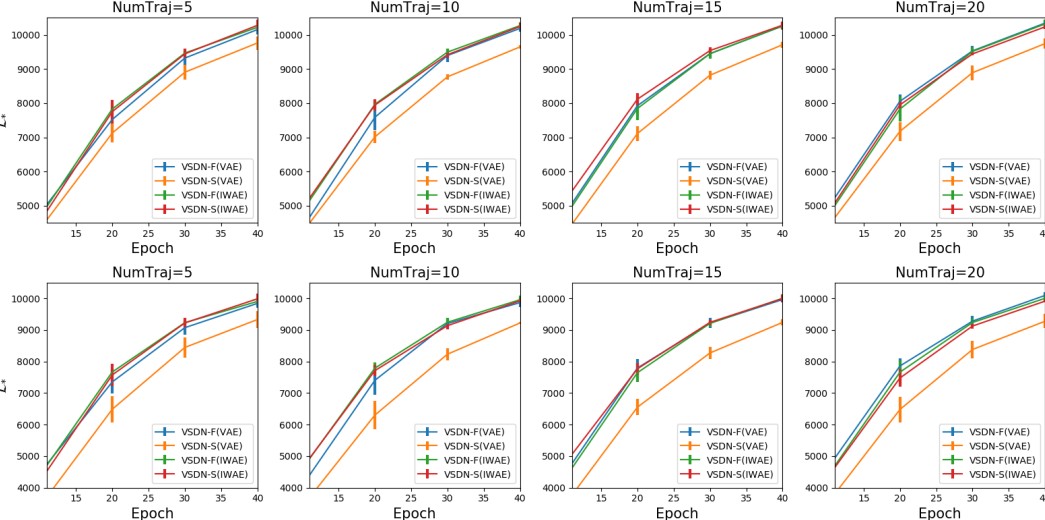

Figure 3: Training processes of our models with respect to the different number of sampled latent state trajectories. (UP: training set; Bottom: validation set)

## 4.3 QUANTITATIVE STUDIES

In order to better understand the properties of VAE and IWAE losses in training VSDNs, we conduct comprehensive quantitative evaluation by varying the number of sampled trajectories when comput-

[‡]https://cdiac.ess-dive.lbl.gov/epubs/ndp/ushcn/monthly_doc.html

ing these losses. We visualize the $\mathcal{L}_{VAE}$ and $\mathcal{L}_{IWAE}^K$ of VSDN trained for $40$ epoches on the Human3.6M dataset in Figure 3. As the VSDN-S contains both forward and backward ODE-RNNs, it is more difficult to train than VSDN-F. The looseness of $\mathcal{L}_{VAE}$ further increases the training difficulty and results in a worse lower bound of VSDN-S (VAE). Therefore, VSDN-S (VAE) requires more epochs to converge during the training. For the other cases, we observe that the $\mathcal{L}_{IWAE}^K$ is tighter than $\mathcal{L}_{VAE}$ in training when the number of trajectories is small. Therefore, $\mathcal{L}_{IWAE}^K$ has faster convergence in training our models. For large number of trajectories, $\mathcal{L}_{VAE}$ has similar tightness as $\mathcal{L}_{IWAE}^K$ in training set.

## 5 CONCLUSIONS

In this paper, we propose a continuous-time stochastic recurrent neural network called VSDN to learn the continuous-time stochastic dynamics from irregular or even sporadic data sequence. We provide two variants, one is VSDN-F whose inference model is a filtering model, and the other is VSDN-S with smoothing inference model. The continuous-time variants of the VAE and IWAE losses are incorporated to efficiently train our model. We demonstrate the effectiveness of VSDN through evaluations studies on different datasets and tasks, and our results show that VSDN can achieve much better performance than state-of-the-art continuous-time deep learning models. In the future work, we will investigate along several potential directions: First, we will apply our models to higher dimensional and more complicated data, such as videos, which are more challenging to model yet, especially under the premise of increasing demand for producing videos in high resolution and frame-per-second (FPS); Second, as stochastic differential equations are the base of many significant control methodologies, we will try to further extend the capacity of our models such that they can be used in precise control scenarios.

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

## A    DEDUCTIONS OF CONTINUOUS-TIME EVIDENCE LOWER BOUND

### A.1    PRELIMINARIES OF STOCHASTIC DIFFERENTIAL EQUATIONS

During the model design and implementation, we will use the Euler–Maruyama method to discretize the stochastic differential equation. The details are given as follows.

**Lemma 1** (Discretization of SDE). *For a SDE $dX = H(X,t)dt + R(t)dW$, we can discretize it as*

$$X_{k+1} = X_k + H(X_k, t_k)\Delta t + R(t_k)\sqrt{\Delta t}\varepsilon, \tag{15}$$

*where $\varepsilon \sim \mathcal{N}(0,1)$, $t_k = k\Delta t$ and $\Delta t$ is the sampling interval. Eq. (15) converges to the original SDE when $\Delta t \to 0$.*

**Lemma 2.** *The state $X_{k+1}$ in Eq.     (15) follows the conditional Gaussian distribution $P(X_{k+1}|X_k) = \mathcal{N}(X_k + H(X_k)\Delta t, \Delta t R(X_k)^T R(X_k))$. The joint distribution of the sate sequence $X_{1:K}$ of Eq. (15) is given by*

$$P(X_{1:K}|X_0) \propto \exp\Big(-0.5\sum_{k=0}^{K-1}(X_{k+1} - m_k)^T \Sigma_k^{-1}(X_{k+1} - m_k)\Big), \tag{16}$$

*where $m_k = X_k + H(X_k)\Delta t$ and $\Sigma_k = \Delta t R(X_k)^T R(X_k)$.*

### A.2    DEVIATION OF $\mathcal{L}_{VAE}$

The proof is similar as the evidence lower bound in (Archambeau et al., 2008). By applying Jensen's inequality, we can obtain that:

$$\log P_{\mathcal{G}}(y_{1:n}) = \log \int P_{\mathcal{G}}(X_{\leq t_n}) \prod_{i=1}^n P_{\mathcal{G}}(y_i|y_{1:n-1}, X_{t_i}) dX_{\leq t_n}$$

$$= \log \int P_{\mathcal{Q}}(X_{\leq t_n}) \frac{P_{\mathcal{G}}(X_{\leq t_n}) \prod_{i=1}^n P_{\mathcal{G}}(y_i|y_{1:n-1}, X_{t_i})}{P_{\mathcal{Q}}(X_{\leq t_n})} dX_{\leq t_n}$$

$$\geq \int P_{\mathcal{Q}}(X_{\leq t_n}) \log \frac{P_{\mathcal{G}}(X_{\leq t_n}) \prod_{i=1}^n P_{\mathcal{G}}(y_i|y_{1:n-1}, X_{t_i})}{P_{\mathcal{Q}}(X_{\leq t_n})} dX_{\leq t_n}$$

$$= \int P_{\mathcal{Q}}(X_{\leq t_n}) \log \frac{P_{\mathcal{G}}(X_{\leq t_n})}{P_{\mathcal{Q}}(X_{\leq t_n})} dX_{t_i} + \int P_{\mathcal{Q}}(X_{\leq t_n}) \log \prod_{i=1}^n P_{\mathcal{G}}(y_i|y_{1:n-1}, X_{t_i}) dX_{\leq t_n}$$

$$= -KL\Big(P_{\mathcal{Q}}||P_{\mathcal{G}}\Big) + \sum_{i=1}^n \mathbb{E}_{P_{\mathcal{Q}}(X_{t_i})} \log P_{\mathcal{G}}(y_i|y_{1:n-1}, X_{t_i}).$$

The next step is to derive the KL divergence term for the prior and inference SDEs. After discretization into $K$ points via Lemma 1, the KL divergence of the two SDEs in VSDN-SDE will be:

$$KL(P_{\mathcal{Q}}||P_{\mathcal{G}}) = \int P_{\mathcal{Q}}(X_{1:K}) \log \frac{P_{\mathcal{Q}}(X_{1:K})}{P_{\mathcal{G}}(X_{1:K})} dX_{1:K}$$

$$= \int \sum_{k=0}^{K-1} P_{\mathcal{Q}}(X_{1:K}) \log \frac{P_{\mathcal{Q}}(X_{k+1}|X_k)}{P_{\mathcal{G}}(X_{k+1}|X_k)} dX_{1:K} = \sum_{k=0}^{K-1} \int P_{\mathcal{Q}}(X_{1:K}) \log \frac{P_{\mathcal{Q}}(X_{k+1}|X_k)}{P_{\mathcal{G}}(X_{k+1}|X_k)} dX_{1:K}$$

$$= \sum_{k=0}^{K-1} \int P_{\mathcal{Q}}(X_{k+2:K}|X_{k+1}) P_{\mathcal{Q}}(X_{k+1}|X_k) P_{\mathcal{Q}}(X_{1:k}) \log \frac{P_{\mathcal{Q}}(X_{k+1}|X_k)}{P_{\mathcal{G}}(X_{k+1}|X_k)} dX_{1:K}$$

$$= \sum_{k=0}^{K-1} \int P_{\mathcal{Q}}(X_{k+1}|X_k) P_{\mathcal{Q}}(X_k) \log \frac{P_{\mathcal{Q}}(X_{k+1}|X_k)}{P_{\mathcal{G}}(X_{k+1}|X_k)} dX_k dX_{k+1}$$

$$= \sum_{k=0}^{K-1} \int P_{\mathcal{Q}}(X_k) \cdot KL\Big(P_{\mathcal{Q}}(X_{k+1}|X_k)||P_{\mathcal{G}}(X_{k+1}|X_k)\Big) dX_k$$

$$= \sum_{k=0}^{K-1} \mathbb{E}_{X_k \sim P_{\mathcal{Q}}(X_k)} KL\Big(P_{\mathcal{Q}}(X_{k+1}|X_k)||P_{\mathcal{G}}(X_{k+1}|X_k)\Big),$$

where $P_{\mathcal{Q}}(X_k)$ is the marginal distribution of $X_k$ in the inference SDE. According to lemma 2 and the KL divergence of two Gaussian distribution, we further have

$$
KL\Big(P_{\mathcal{Q}}(X_{k+1}|X_k)||P_{\mathcal{G}}(X_{k+1}|X_k)\Big) = \frac{1}{2}\big(tr(\Sigma_{k,\mathcal{G}}^{-1}\Sigma_{k,\mathcal{Q}}) + (m_{k,\mathcal{G}} - m_{k,\mathcal{Q}})^T\Sigma_{k,\mathcal{G}}^{-1}(m_{k,\mathcal{G}} - m_{k,\mathcal{Q}})
$$

$$
+ \log\frac{\det\Sigma_{k,\mathcal{G}}}{\det\Sigma_{k,\mathcal{Q}}} - d\Big)
$$

$$
= \frac{1}{2}\Big(tr\Big((R_{\mathcal{G}}R_{\mathcal{G}}^T)^{-1}R_{\mathcal{Q}}R_{\mathcal{Q}}^T\Big) + {\color{red}\Delta t(H_{\mathcal{G}} - H_{\mathcal{Q}})^T(R_{\mathcal{G}}R_{\mathcal{G}}^T)^{-1}(H_{\mathcal{G}} - H_{\mathcal{Q}})} + \log\frac{R_{\mathcal{G}}R_{\mathcal{G}}^T}{R_{\mathcal{Q}}R_{\mathcal{Q}}^T} - d\Big)
$$

where $d$ is the dimension of $X_{k+1}$. When we restrict $R_{\mathcal{G}} = R_{\mathcal{Q}}$, we have

$$
KL(P_{\mathcal{Q}}||P_{\mathcal{G}}) = \frac{1}{2}\sum_{k=0}^{K-1}\mathbb{E}_{X_k\sim P_{\mathcal{Q}}(X_k)}(H_{\mathcal{G}} - H_{\mathcal{Q}})^T(R_{\mathcal{G}}R_{\mathcal{G}}^T)^{-1}(H_{\mathcal{G}} - H_{\mathcal{Q}})\Delta t.
$$

When we set $\Delta t \to 0$, the discretized SDEs converge to the original SDEs and $KL(P_{\mathcal{Q}}||P_{\mathcal{G}})$ converges to:

$$
KL(P_{\mathcal{Q}}||P_{\mathcal{G}}) = \frac{1}{2}\int_0^{t_n}\mathbb{E}_{P_{\mathcal{Q}}(X_t)}\Big((H_{\mathcal{Q}} - H_{\mathcal{G}})^T[R_{\mathcal{G}}R_{\mathcal{G}}^T]^{-1}(H_{\mathcal{Q}} - H_{\mathcal{G}})\Big)dt.
$$

The expectation operator is removed as $H_{\mathcal{G}}$, $H_{\mathcal{Q}}$ and $R_{\mathcal{G}}$ are independent with $X_t$.

If $R_{\mathcal{G}}$ does not equal to $R_{\mathcal{Q}}$, we have

$$
KL(P_{\mathcal{Q}}||P_{\mathcal{G}}) = \frac{1}{2}\lim_{\Delta t\to 0}\sum_{k=0}^{K-1}\mathbb{E}_{X_k\sim P_{\mathcal{Q}}(X_k)}\Big((H_{\mathcal{G}} - H_{\mathcal{Q}})^T(R_{\mathcal{G}}R_{\mathcal{G}}^T)^{-1}(H_{\mathcal{G}} - H_{\mathcal{Q}}) + \frac{const}{\Delta t}\Big)\Delta t,
$$

$$
= +\infty
$$

### A.3 Deviation of $\mathcal{L}_{IWAE}$

Given $X_{k+1} = X_k + H_{\mathcal{Q}}\Delta t + R_{\mathcal{G}}\sqrt{\Delta t}\varepsilon = m_{k,\mathcal{Q}} + R_{\mathcal{G}}\sqrt{\Delta t}\varepsilon$, we have:

$$
\log w = \log\frac{P_{\mathcal{G}}(x_{\leq t_n})}{P_{\mathcal{Q}}(x_{\leq t_n})} = \sum_{k=0}^{K-1}\log\frac{P_{\mathcal{G}}(X_{k+1}|X_k)}{P_{\mathcal{Q}}(X_{k+1}|X_k)}
$$

$$
= \frac{1}{2}\sum_{k=0}^{K-1} -(X_{k+1} - m_{k,\mathcal{G}})^T\Sigma_{k,\mathcal{G}}^{-1}(X_{k+1} - m_{k,\mathcal{G}}) + (X_{k+1} - m_{k,\mathcal{Q}})^T\Sigma_{k,\mathcal{G}}^{-1}(X_{k+1} - m_{k,\mathcal{Q}})
$$

$$
= \frac{1}{2}\sum_{k=0}^{K-1} -\Big((H_{\mathcal{Q}} - H_{\mathcal{G}})\Delta t + R_{\mathcal{G}}\sqrt{\Delta t}\varepsilon\Big)^T[\Delta t R_{\mathcal{G}}R_{\mathcal{G}}^T]^{-1}\Big((H_{\mathcal{Q}} - H_{\mathcal{G}})\Delta t + R_{\mathcal{G}}\sqrt{\Delta t}\varepsilon\Big)
$$

$$
+ \Big(R_{\mathcal{G}}\sqrt{\Delta t}\varepsilon\Big)^T[\Delta t R_{\mathcal{G}}R_{\mathcal{G}}^T]^{-1}\Big(R_{\mathcal{G}}\sqrt{\Delta t}\varepsilon\Big)
$$

$$
= \frac{1}{2}\sum_{k=0}^{K-1} -(H_{\mathcal{Q}} - H_{\mathcal{G}})^T[R_{\mathcal{G}}R_{\mathcal{G}}^T]^{-1}(H_{\mathcal{Q}} - H_{\mathcal{G}})\Delta t - 2(H_{\mathcal{Q}} - H_{\mathcal{G}})^T[R_{\mathcal{G}}]^{-1}\sqrt{\Delta t}\varepsilon \quad (17)
$$

Let $\Delta t \to 0$, we have

$$
\log w = \frac{1}{2}\int -(H_{\mathcal{Q}} - H_{\mathcal{G}})^T[R_{\mathcal{G}}R_{\mathcal{G}}^T]^{-1}(H_{\mathcal{Q}} - H_{\mathcal{G}})dt - \int(H_{\mathcal{Q}} - H_{\mathcal{G}})^T[R_{\mathcal{G}}]^{-1}dW_t, \quad (18)
$$

which is equivalent to

$$
d\log w = -(H_{\mathcal{Q}} - H_{\mathcal{G}})^T[2R_{\mathcal{G}}R_{\mathcal{G}}^T]^{-1}(H_{\mathcal{Q}} - H_{\mathcal{G}})dt - (H_{\mathcal{Q}} - H_{\mathcal{G}})^T[R_{\mathcal{G}}]^{-1}dW_t. \quad (19)
$$

## B Illustration of the Noise Injection of $R(X_t)$

In the section, we give an example to illustrate the noise injection problem when we include $X_t$ as the input for the diffusion function $R(X_t)$ in a Neural SDE. For simplicity, we consider the scalar case (i.e. $X_t \in \mathbb{R}$).

### B.1 Case A: $R$ is independent of $X_t$

Consider the following neural SDE:

$$dX_t = H_\phi(X_t; t)dt + R_\theta(t)dW_t, \tag{20}$$

where $H_\phi$ and $R_\theta$ are neural networks. $\phi$ denotes the parameters of the drift network and $\theta$ denotes the parameters of the diffusion network.

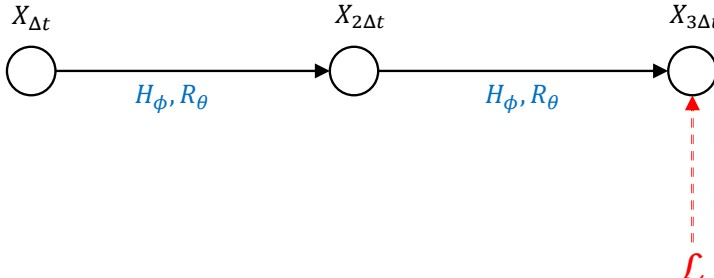

Figure 4: A example to show the noise injection problem of $R(X_t)$.

Now consider the following example (shown in Figure 4) that we have to compute the gradient of the loss at $t = 3\Delta t$ with respect to the network parameters, where the neural SDE is discretized by Euler–Maruyama method:

$$X_{\Delta t} = X_0 + H_\phi(X_0; 0)\Delta t + R_\theta(0)\sqrt{\Delta t}\varepsilon_1, \tag{21}$$

$$X_{2\Delta t} = X_{\Delta t} + H_\phi(X_{\Delta t}; \Delta t)\Delta t + R_\theta(\Delta t)\sqrt{\Delta t}\varepsilon_2, \tag{22}$$

$$X_{3\Delta t} = X_{2\Delta t} + H_\phi(X_{2\Delta t}; 2\Delta t)\Delta t + R_\theta(2\Delta t)\sqrt{\Delta t}\varepsilon_3, \tag{23}$$

where $\varepsilon_n \sim \mathcal{N}(0, 1)$. It is straight forward to prove the following lemma. For notation simplicity, we define $H_\phi(n) = H_\phi(X_{(n-1)\Delta t}; (n-1)\Delta t)$ and $R_\theta(n) = R_\theta((n-1)\Delta t)$.

**Lemma 3.** *Eqs. (21) − (23) follows the following relationship of the gradients:*

$$\frac{\partial X_{n\Delta t}}{\partial X_{(n-1)\Delta t}} = 1 + \Delta t\frac{\partial H_\phi(n)}{\partial X_{(n-1)\Delta t}} \tag{24}$$

Therefore, the gradients of the parameters in the drift and diffusion functions can be given by:

$$\begin{aligned}
\frac{\partial \mathcal{L}}{\partial \phi} =& \frac{\partial \mathcal{L}}{\partial X_{3\Delta t}}\frac{\partial X_{3\Delta t}}{\partial \phi} + \frac{\partial \mathcal{L}}{\partial X_{3\Delta t}}\frac{\partial X_{3\Delta t}}{\partial X_{2\Delta t}}\frac{\partial X_{2\Delta t}}{\partial \phi} + \frac{\partial \mathcal{L}}{\partial X_{3\Delta t}}\frac{\partial X_{3\Delta t}}{\partial X_{2\Delta t}}\frac{\partial X_{2\Delta t}}{\partial X_{\Delta t}}\frac{\partial X_{\Delta t}}{\partial \phi}, \\
=& \frac{\partial \mathcal{L}}{\partial X_{3\Delta t}}\frac{\partial H_\phi(3)}{\partial \phi}\Delta t + \frac{\partial \mathcal{L}}{\partial X_{3\Delta t}}\Big[1 + \Delta t\frac{\partial H_\phi(3)}{\partial X_{2\Delta t}}\Big]\frac{\partial H_\phi(2)}{\partial \phi}\Delta t \\
&+ \frac{\partial \mathcal{L}}{\partial X_{3\Delta t}}\Big[1 + \Delta t\frac{\partial H_\phi(3)}{\partial X_{2\Delta t}}\Big]\Big[1 + \Delta t\frac{\partial H_\phi(2)}{\partial X_{\Delta t}}\Big]\frac{\partial H_\phi(1)}{\partial \phi}\Delta t.
\end{aligned} \tag{25}$$

and

$$\begin{aligned}
\frac{\partial \mathcal{L}}{\partial \theta} =& \frac{\partial \mathcal{L}}{\partial X_{3\Delta t}}\frac{\partial X_{3\Delta t}}{\partial \theta} + \frac{\partial \mathcal{L}}{\partial X_{3\Delta t}}\frac{\partial X_{3\Delta t}}{\partial X_{2\Delta t}}\frac{\partial X_{2\Delta t}}{\partial \theta} + \frac{\partial \mathcal{L}}{\partial X_{3\Delta t}}\frac{\partial X_{3\Delta t}}{\partial X_{2\Delta t}}\frac{\partial X_{2\Delta t}}{\partial X_{\Delta t}}\frac{\partial X_{\Delta t}}{\partial \theta}, \\
=& \frac{\partial \mathcal{L}}{\partial X_{3\Delta t}}\frac{\partial R_\theta(3)}{\partial \theta}\sqrt{\Delta t}\varepsilon_3 + \frac{\partial \mathcal{L}}{\partial X_{3\Delta t}}\Big[1 + \Delta t\frac{\partial H_\phi(3)}{\partial X_{2\Delta t}}\Big]\frac{\partial R_\theta(2)}{\partial \theta}\sqrt{\Delta t}\varepsilon_2 \\
&+ \frac{\partial \mathcal{L}}{\partial X_{3\Delta t}}\Big[1 + \Delta t\frac{\partial H_\phi(3)}{\partial X_{2\Delta t}}\Big]\Big[1 + \Delta t\frac{\partial H_\phi(2)}{\partial X_{\Delta t}}\Big]\frac{\partial R_\theta(1)}{\partial \theta}\sqrt{\Delta t}\varepsilon_1.
\end{aligned} \tag{26}$$

According to Eq. (25) and Eq. (26), the gradient of $\phi$ of the drift network is deterministic except the weight of first hidden layer and the gradient of $\theta$ of the diffusion network is obstructed by Gaussian noise terms $\sqrt{\Delta t}\varepsilon_1$, $\sqrt{\Delta t}\varepsilon_2$ and $\sqrt{\Delta t}\varepsilon_3$.

## B.2 CASE B: $R$ USES $X_t$ AS INPUT

Now we consider the case when the diffusion network $R$ also use $X_t$ as input. Eq. (24) will change to the following equation:

$$\frac{\partial X_{n\Delta t}}{\partial X_{(n-1)\Delta t}} = 1 + \Delta t \frac{\partial H_\phi(n)}{\partial X_{(n-1)\Delta t}} + \sqrt{\Delta t}\varepsilon_n \frac{\partial R_\theta(n)}{\partial X_{(n-1)\Delta t}}. \tag{27}$$

Inserting Eq. (27) into Eq. (25) and Eq. (26), we have

$$\begin{aligned}
\frac{\partial \mathcal{L}}{\partial \phi} =& \frac{\partial \mathcal{L}}{\partial X_{3\Delta t}} \frac{\partial X_{3\Delta t}}{\partial \phi} + \frac{\partial \mathcal{L}}{\partial X_{3\Delta t}} \frac{\partial X_{3\Delta t}}{\partial X_{2\Delta t}} \frac{\partial X_{2\Delta t}}{\partial \phi} + \frac{\partial \mathcal{L}}{\partial X_{3\Delta t}} \frac{\partial X_{3\Delta t}}{\partial X_{2\Delta t}} \frac{\partial X_{2\Delta t}}{\partial X_{\Delta t}} \frac{\partial X_{\Delta t}}{\partial \phi}, \\
=& \frac{\partial \mathcal{L}}{\partial X_{3\Delta t}} \frac{\partial H_\phi(3)}{\partial \phi}\Delta t + \frac{\partial \mathcal{L}}{\partial X_{3\Delta t}} \Big[ 1 + \Delta t \frac{\partial H_\phi(3)}{\partial X_{2\Delta t}} + \sqrt{\Delta t}\varepsilon_3 \frac{\partial R_\theta(3)}{\partial X_{2\Delta t}} \Big] \frac{\partial H_\phi(2)}{\partial \phi}\Delta t \\
&+ \frac{\partial \mathcal{L}}{\partial X_{3\Delta t}} \Big[ 1 + \Delta t \frac{\partial H_\phi(3)}{\partial X_{2\Delta t}} + \sqrt{\Delta t}\varepsilon_3 \frac{\partial R_\theta(3)}{\partial X_{2\Delta t}} \Big] \Big[ 1 + \Delta t \frac{\partial H_\phi(2)}{\partial X_{\Delta t}} + \sqrt{\Delta t}\varepsilon_2 \frac{\partial R_\theta(2)}{\partial X_{\Delta t}} \Big] \frac{\partial H_\phi(1)}{\partial \phi}\Delta t.
\end{aligned} \tag{28}$$

and

$$\begin{aligned}
\frac{\partial \mathcal{L}}{\partial \theta} =& \frac{\partial \mathcal{L}}{\partial X_{3\Delta t}} \frac{\partial X_{3\Delta t}}{\partial \theta} + \frac{\partial \mathcal{L}}{\partial X_{3\Delta t}} \frac{\partial X_{3\Delta t}}{\partial X_{2\Delta t}} \frac{\partial X_{2\Delta t}}{\partial \theta} + \frac{\partial \mathcal{L}}{\partial X_{3\Delta t}} \frac{\partial X_{3\Delta t}}{\partial X_{2\Delta t}} \frac{\partial X_{2\Delta t}}{\partial X_{\Delta t}} \frac{\partial X_{\Delta t}}{\partial \theta}, \\
=& \frac{\partial \mathcal{L}}{\partial X_{3\Delta t}} \frac{\partial R_\theta(3)}{\partial \theta}\sqrt{\Delta t}\varepsilon_3 + \frac{\partial \mathcal{L}}{\partial X_{3\Delta t}} \Big[ 1 + \Delta t \frac{\partial H_\phi(3)}{\partial X_{2\Delta t}} + \sqrt{\Delta t}\varepsilon_3 \frac{\partial R_\theta(3)}{\partial X_{2\Delta t}} \Big] \frac{\partial R_\theta(2)}{\partial \theta}\sqrt{\Delta t}\varepsilon_2 \\
&+ \frac{\partial \mathcal{L}}{\partial X_{3\Delta t}} \Big[ 1 + \Delta t \frac{\partial H_\phi(3)}{\partial X_{2\Delta t}} + \sqrt{\Delta t}\varepsilon_3 \frac{\partial R_\theta(3)}{\partial X_{2\Delta t}} \Big] \Big[ 1 + \Delta t \frac{\partial H_\phi(2)}{\partial X_{\Delta t}} + \sqrt{\Delta t}\varepsilon_2 \frac{\partial R_\theta(2)}{\partial X_{\Delta t}} \Big] \frac{\partial R_\theta(1)}{\partial \theta}\sqrt{\Delta t}\varepsilon_1.
\end{aligned} \tag{29}$$

According to Eq. (28), the gradient of $\phi$ is now also corrupted by noise terms (i.e. $\sqrt{\Delta t}\varepsilon_3$ and $\Delta t\varepsilon_2\varepsilon_3$), while in previous case it is deterministic. What's worse, more noise terms are added into the gradient of $\theta$. When we train our models in long data sequence, these injected noise terms will cause a large variance of the parameters' gradients. Therefore, we can conclude that introducing $X_t$ into the diffusion function is not beneficial.

## C MODEL CONFIGURATION

### C.1 HUMAN MOTION ACTIVITIES

For all the models, the feed-forward network contains one hidden layer with 256 Relu units. $\Delta t$ is set as 0.25. The dimension of hidden features of ODE-RNN and GRU-ODE is 512 and the dimension of latent states is 128. A single-layer feed-forward network with 128 Relu units is defined to compute the initial states of the latent state. For LatentSDE, the posterior initial state is computed by using the encoding feature of a backward ODE-RNN. The number of latent state trajectories generated to compute VAE and IWAE losses is 5.

All models are trained by Adam optimizer with learning rate 0.0001 and weight-decay 0.0005. The batch size is 64. Early stopping with 10 epoch tolerance is applied.

### C.2 TOY SIMULATION AND CLIMATE PREDICTION

For all the models, the feed-forward network contains one hidden layer with 25 Relu units. $\Delta t$ is set as 0.1 for USHCN and 0.01 for Double-OU. The dimension of hidden features of ODE-RNN and GRU-ODE is 15 and the dimension of latent states is 15 as well. A single-layer feed-forward network with 128 Relu units is defined to compute the initial states of the latent state. The number of latent state trajectories generated to compute VAE and IWAE losses is 5.

All models are trained by Adam optimizer with learning rate 0.0001 and weight-decay 0.0001. The batch size is 500 for USHCN and 250 for Double-OU. Early stopping with 25 epoch tolerance is applied.

### C.3 SPECTROGRAM MODELING

To further evaluate the performance of our model in high-dimensional data, we conduct a brief experiment on the spectrogram data extracted from the FMA dataset (Defferrard et al., 2017), which is a large collection of music and songs. We transform the first 500 songs in FMA-small into spectrogram and then split the spectrogram into segments. Each segment has 100 frames and each frame is 1025 dimensional. Each dimension of the data frame corresponds to a specific frequency component of STFT. The definitions of prediction and interpolation tasks are same as those in Section 4.1.

Table 4: Model performance on Spectrogram dataset

|  | Prediction | | Interpolation | |
| --- | --- | --- | --- | --- |
|  | NLL | MSE | NLL | MSE |
| LatentODE | −0.1356 | 0.066 | −0.1369 | 0.066 |
| LatentSDE | −0.1403 | 0.065 | −0.1291 | 0.066 |
| GRU-ODE | −0.1610 | **0.058** | −0.1603 | 0.059 |
| ODE-RNN | −0.1677 | **0.058** | −0.1683 | **0.058** |
| **VSDN-F (VAE)** | −0.1969 | 0.059 | −0.1947 | 0.059 |
| **VSDN-S (VAE)** | −0.1994 | 0.059 | −0.1981 | 0.059 |
| **VSDN-F (IWAE)** | −0.1931 | 0.059 | −0.1935 | 0.059 |
| **VSDN-S (IWAE)** | **−0.2020** | 0.059 | **−0.2017** | 0.059 |

For all the models, the feed-forward network contains one hidden layer with 256 Relu units. The dimension of hidden features of ODE-RNN and GRU-ODE is 256 and the dimension of latent states is 64. A single-layer feed-forward network with 128 Relu units is defined to compute the initial states of the latent state. For LatentSDE, the posterior initial state is computed by using the encoding feature of a backward ODE-RNN. The number of latent state trajectories generated to compute VAE and IWAE losses is 5. The batch size is 32.

The NLL and MSE (per dim) are shown in Table 4. Our model has similar mean square errors with baselines but has much better NLL, which indicates that our model can better estimate the stochastic process of the data.

