# OpenReview forum: "Learning Continuous-Time Dynamics by Stochastic Differential Networks"
_ICLR.cc/2021/Conference — Reject_

### Official Review · AnonReviewer4 · 2020-10-25
**Need to better understand different aspects of the paper**

**Rating:** 5
**Confidence:** 3

**Review:**

In this paper the author/s study/ies the fundamental problem of learning continuous-time stochastic dynamics, in the case where the available time series data suffer irregularity and sparseness. The paper assumes that high dimensional data are generated from a system where latent states are observed. The paper tackles the problem
In such a settings, it is generally impossible to derive a continuous-time stochastic process which precisely describes the many behaviors of the system under study.
The paper develops a method base on Variational Bayes applied to learn flexible continuous-time stochastic recurrent neural network, they call Variational Stochastic Differential Networks. The particualr feature of such model is to capture the stochastic dependency among latent states and observations.
The paper provides theoretichal tool, under the form of lower bounds for the efficient training of the neural model.
THe author/s state that the results from a rich set of numerical experiments witnesses that the proposed approach
outperforms state-of-the-art continuous-time deep learning models to solve the prediction and interpolation tasks, in the particular case when irrelugar and sporading time series data is available.
----------------------------------------------------------------------------------------------------------------------

Reason for Score:
Overall, I vote for rejecting. I like the idea to mix parametric and non parametric components to learn time series dynamics. However, I have many concerns that are made explicit in the cons section.
Hopefully the authors can address my concerns in the rebuttal period.

----------------------------------------------------------------------------------------------------------------------
Pros.
1) the paper tackles a relevant problem.
2) I appreciate the idea to combine parametric and non parametric models
3) the proposed model is clear
3) numerical experiments, on the selected data sets, witness in favour of the proposed method
----------------------------------------------------------------------------------------------------------------------

Cons.
1) why other models for time series modeling and forecasting have not been taken into accout? what about dynamic Bayesian networks, hidden Markov models, contiuous time Bayesian networks, and many other ...

2) the paper states the interest is related to high dimensional time series, but the data sets used for numerical experiments, in my humble opinion, are not as such, they are of very small or moderate dimensionality, in terms of number of variables.

3) I found no discussion about the choice of the dimension of the latent space, maybe my fauls and if this is the case I apologize, but still I think that more attention and more motivations must be given in the numerical experiments section.

4) I'm under the impression that the proposed solution is a sensible combination of existing results and thus the work is somewhat of incremental nature, which I do not know whether is relevant for ICLR.

5) I'm confused on the formulation of the inference problems, the paper considers both filtering and smoothing. However, later in the paper, the numerical experiments section, I read about "prediction" and "interpolation" which in my uderstanding is the same as filtering and smoothing. Thus, I would like the authors to better frame their problem.

6) at page 2, I can not understand the difference between path and history, if they are used as synonims, please do not do it, it increases confusion to the interested reader, try to be more formal and clear as possible to let interested reader to appreciate your contributions.

7) at page 3 I read "However, RG only use the historical data as input, as we observe no improvement by including the
current state.", I would ask to motivate, personally I'm not that surprised by this but I would like the paper to help understand the why of this occuring.

8) In equation (5) you are using the entire trajectory and thus you are performing smoothing, I think the problem/s tackled must be clearly described.

9) at page 5 I read "In real-world applications, the latent state may not have strong dependency on future observations". This is somewhat surprising me, which has not strong dependency on which? the latent does not influence the future? the future does not tell much about latent? This sentence in obscure to my elementary mind, and thus I kindly ask to better clarify on it.

10) Table 1 and 2, please call the macro column filtering and smoothing in place of prediction and interpolation, maybe I completely miss the meaning of the difference you made between prediction and filtering and the difference between smoothing and interpolation. If this is the case, please better clarify on these aspects.

11) NLL and MSE of VSDN-S (VAE) are always better that those achieved by VSDN-F (VAE), this is not that strange, one is using more info than the other, as well as I understood. The same does not apply to VSDN-S (IWAE) and VSDN-F (IWAE), and this is surprising in my humble opinion, maybe not found the optimum?

12) In table 2, NLL is positive for LatentSDE, can you comment on this?

13) The same as 12) in Table 3, I kindly ask to clarify under which circumstances this happens.
----------------------------------------------------------------------------------------------------------------------
Questions during rebuttal period:

Please address and clarify the cons above
----------------------------------------------------------------------------------------------------------------------
Some typos:

page 3: I read "However, RG only use the historical", it should read "However, RG only uses the historical"
        I read "HQ also use the", it should read "HQ also uses the"
----------------------------------------------------------------------------------------------------------------------

---

> ### Author Response · Authors · 2020-11-23
> **Response to AnonReviewer4 (Part 1/3)**
>
> **Q1: why other models for time series modeling and forecasting have not been taken into account? what about dynamic Bayesian networks, hidden Markov models, continuous time Bayesian networks, and many other ...**
>
> ---- There are rich literature works showing that deep neural net based models outperform the traditional probabilistic graphical models (e.g., HMM, DBN) in many applications. Therefore, we focus more on the comparison with the deep neural net based models. Some references are given as follows:
>
> HMM:
>
> 1. Nicolas Boulanger-Lewandowski, Yoshua Bengio, and Pascal Vincent. "Modeling temporal dependencies in high-dimensional sequences: application to polyphonic music generation and transcription." In Proceedings of the 29th International Conference on International Conference on Machine Learning (ICML'12). Omnipress, Madison, WI, USA, 1881–1888.
>
> 2. Lefebvre,Grégoire, et al. "BLSTM-RNN based 3D gesture classification." International conference on artificial neural networks. Springer, Berlin, Heidelberg, 2013.
>
> 3. Rajib Ghosh, Chirumavila Vamshi, Prabhat Kumar, "RNN based online handwritten word recognition in Devanagari and Bengali scripts using horizontal zoning",  Pattern Recognition, Volume 92, 2019, Pages 203-218.
>
> 4. Deshmukh, Akshay Madhav. "Comparison of Hidden Markov Model and Recurrent Neural Network in Automatic Speech Recognition." European Journal of Engineering Research and Science 5, no. 8 (2020): 958-965.
>
> DBN:
>
> 1. Wöllmer, M., Eyben, F., Graves, A., Schuller, B., & Rigoll, G. "Bidirectional LSTM networks for context-sensitive keyword detection in a cognitive virtual agent framework." Cognitive Computation, 2(3), 180-190.
>
> 2. Nguyen, Duc-Canh, Gérard Bailly, and Frédéric Elisei. "Learning off-line vs. on-line models of interactive multimodal behaviors with recurrent neural networks." Pattern Recognition Letters 100 (2017): 29-36.
>
> **Q2: the paper states the interest is related to high dimensional time series, but the data sets used for numerical experiments, in my humble opinion, are not as such, they are of very small or moderate dimensionality, in terms of number of variables.**
>
> ----  In our motion experiments, the dimension of each data frame is 62 in MoCap dataset and 51 in h36m dataset. The dimensionalities of the data in our experiments are similar as those in the literature works on deep generative models of sporadic data [1 - 4].
> 1. Wei Cao, Dong Wang, Jian Li, Hao Zhou, Yitan Li, and Lei Li. 2018. "BRITS: bidirectional recurrent imputation for time series." In Proceedings of the 32nd International Conference on Neural Information Processing Systems (NIPS'18). 6776–6786.
> 2. Yonghong Luo, Xiangrui Cai, Ying Zhang, Jun Xu, and Xiaojie Yuan. 2018. "Multivariate time series imputation with generative adversarial networks." In Proceedings of the 32nd International Conference on Neural Information Processing Systems (NIPS'18). 1603–1614.
> 3. Yulia Rubanova, Tian Qi Chen, and David K Duvenaud.  "Latent ordinary differential equations for irregularly-sampled time series."  InAdvances in Neural Information Processing Systems 32, pp.5321–5331. 2019.
> 4. Edward De Brouwer, Jaak Simm, Adam Arany, and Yves Moreau.  "GRU-ODE-Bayes: Continuous Modeling of sporadically-observed time series."   InAdvances in Neural Information ProcessingSystems 32, pp. 7379–7390. 2019.
>
> We also run an experiment on high-dimensional (above 1k) data. Our model has similar mean square errors with baselines but has much better NLL, which indicates that our model can better estimate the stochastic process of the data. In our future works, we will extend our work on more challenging very high-dimensional data (e.g. video).
>
> **Q3: I found no discussion about the choice of the dimension of the latent space**
>
> ---- The detailed model configuration of our experiments is given in Appendix C. In our preliminary studies, we tried different number of dimensions of the latent state. When the number of the dimensions of the latent state is too small, the performance of our model will degenerate. With large dimensions of the latent state, the performance is not sensitive with respect to the dimensions of latent state.

---

> > ### Comment · AnonReviewer4 · 2020-11-23
> > **Thx for providing clarifications**
> >
> > Your answer to Q1: I think this is not enough to exclude other competing models, the literature witnesses that in some application domains neural networks work fine. However, unless, the value of what you proposed is strictly confined to the specific application, which I do think is not the case, I still think other methods, model based could be interesting to be taken into account.
> >
> > Your answer to Q2: I think that I miss something because I ddi not find the additional experiments of 1k dimensions, so please help me to understand where I'm wrong.
> >
> > You answer to Q3: the result of such an insensitive performance with respect to the dimension of the latent space is very curious.

---

> > > ### Author Response · Authors · 2020-11-25
> > > **Additional Experiments**
> > >
> > > **Q2: I did not find the additional experiments of 1k dimensions, so please help me to understand where I'm wrong.**
> > >
> > > ---- We have submitted a rebuttal update version into the system. The additional experiments on high-dimensional data is in Appendix C.3.

---

> ### Author Response · Authors · 2020-11-23
> **Response to AnonReviewer4 (Part 2/3)**
>
> **Q5: I'm confused on the formulation of the inference problems**
>
> **Q8: I think the problem/s tackled must be clearly described.**
>
> **Q10: difference between prediction and filtering,  and the difference between smoothing and interpolation.**
>
> ----  Thank you for pointing out the misleading part. We propose a continuous-time stochastic recurrent neural network as the generative model. In order to train the generative model, we introduce the inference model and train both of them by the evidence lower bound. We evaluate the performance of the generative model (i.e. only the generative model is considered in the prediction and interpolation tasks) because the motivation of this paper is to propose an accurate continuous-time online predictive model.
>
> ----  The definitions of prediction and interpolation are not the same as filtering and smoothing.  In all our experiments, some points of the data sequence are used as observed data (which are used to train the model), while others are considered as the missing data that we have to impute. To evaluate the model performance, we define the prediction task as to predict the sequence of the observed data, and the interpolation task as to approximate the missing data. In all these tasks, our models only use the generative model to recurrently predict the future values, while filtering/smoothing are inference models that are used to help train the generative ones.
>
> ----  The definitions of prediction and interpolation are originally given in Section 4.1 on Page 5. We will claim it more clearly. Besides, we will add a discussion in Section 3 to clearly describe the functionality of each component.
>
>  >“
> Applications: VSDN consists of a generative model and an inference model. The generative model is an online predictive model which can recurrently predict the future values of the sequence.  The Inference models can be applied to either filtering or smoothing problems of the latent states accordingly. Furthermore, the smoothing inference model infers the latent state trajectory from the whole sequence, which can be further used in Eq. (3) to synthesize missing data. Therefore, the smoothing inference model is capable of offline interpolation. The motivation of this paper is to design an efficient continuous-time stochastic recurrent model. Therefore, VSDNs only use the generative model to recurrently predict the future values in the experiments.
> ”
>
> **Q7: RG only use the historical data as input**
>
> ---- It is not beneficial to include $X_t$ as the input of the diffusion network $R_\mathcal{G}$ because that will introduce additional variance into the gradients of the model parameters. In computing VAE and IWAE losses, it is required to generate samples of latent state from the stochastic differential equation and introduce Gaussian noise in the samples. Adding $X_t$ into $R_\mathcal{G}$ will inject more noisy terms when computing the gradients of the parameters during the training.
> In order to better describe this noise injection problem, we have added an example in Appendix B and explicitly derived the closed-form solution of the parameters’ gradients, from which we can make the above conclusion.
>
> **Q9: the latent state may not have strong dependency on future observations**
>
> ----  Thank you for pointing out the misleading part. In principle, the inference modal should approximate the exact posterior distribution $P(X_{\leq t_N}|y_{1:N})$. At each time $t$, we should approximate $P(X_{t}|X_{<t}, y_{1:N})$, where $y_{1:N}$ is the whole observation sequence.  Although future observations do tell some information of the latent state, we can approximate  $P(X_{t}|X_{<t}, y_{1:N})$ by $P(X_{t}|X_{<t}, y_{t_n\leq t})$  where $ y_{t_n\leq t} $ is the observations up till current time, and get close results (as shown in the experiments of double-OU in Table 3). Instead, VSDN-F does not have to include a backward RNN to process the future observations and thus its running speed is faster. We will claim our motivation of filtering inference model more clearly in the revision. In the updated version, the original sentence is replaced with
>
> > "In real-world applications, it is sometimes possible to have close performance in inference with-out processing the future observations."
>
> and at the end of the paragraph, we add
>
> > "The filtering model does not have to include a backward RNN to process the future observa-tions and thus its running speed is faster."

---

> ### Author Response · Authors · 2020-11-23
> **Response to AnonReviewer4 (Part 3/3)**
>
> **Q11: VSDN-S (IWAE) is slightly worse than VSDN-F (IWAE)**
>
> ---- In most cases, the VSDN-S(IWAE) has better NLL than VSDN-F(IWAE) (e.g. −127 vs −125 in Table 1, −115 vs −110 in Table 2, 0.654 vs 0.661 in Table 3). The results of NLL are consistent with literature that IWAE generally has better log-likelihood than VAE [1][2]. We do observe that VSDN-S(IWAE) has slightly worse VSDN-F(IWAE) in some cases (e.g. 0.1797 vs 0.1751 in Human3.6M dataset (Table 1), 0.381 vs 0.370 in USHCN(Table 3)). This may be caused by randomness of the training process. The other possible reason is the noise in the gradient of the inference model. In [2], it is shown that IWAE can be detrimental to the learning of an inference network by reducing the signal-to-noise ratio of the gradient estimator.
>
> 1. Burda, Yuri, Roger Grosse, and Ruslan Salakhutdinov. "Importance weighted autoencoders." arXiv preprint arXiv:1509.00519 (2015).
>
> 2. Rainforth, Tom, Adam R. Kosiorek, Tuan Anh Le, Chris J. Maddison, Maximilian Igl, Frank Wood, and Yee Whye Teh. "Tighter variational bounds are not necessarily better." ICML (2018).
>
>
> **Q12&13: Some NLLs of LatentSDE are positive.**
>
> ---- The negative log-likelihood (NLL) indicates whether the learned model fits the distribution of data. For discrete-value data, NLL is always positive. For continuous-value data (as is the data we used in our experiment), NLL can be negative when the learned probabilistic density P(x) > 1 for the data. Smaller NLL is better. In the original submission, a positive NLL simply means that the model has worse performance in the experiments. We have already double-checked the architecture of LatentSDE and re-evaluated the model performance. In our updated results, LatentSDEs also have negative NLLs in Table 2.

---

### Official Review · AnonReviewer2 · 2020-10-27
**This paper proposes a new continuous-time stochastic recurrent network called Variational Stochastic Differential Network (VSDN) that incorporates Stochastic Differential Equations (SDEs) into recurrent neural model to effectively model the continuous-time stochastic dynamics based only on sparse or irregular observations.**

**Rating:** 7
**Confidence:** 4

**Review:**

This paper aims to model the complicated continuous time-series by using SDE for modeling the latent state trajectories. The authors claim that using SDE instead of ODE for the latent states has higher flexibility to capture more complex dynamics. Then they propose a continuous-time versions of variational evidence lower bounds (ELBO) which can be trained using ODE-RNNs as the inference networks. The proposed VSDN model has the capability to capture the latent state stochasticity not via the initial states but rather via SDEs, as opposed to other methods like latent ODE and latent SDE.

Overall, the paper is interesting as it employs many recent advances in the field to introduce a richer model for the continuous complex time-series with the capability of taking into account the stochasticity of the latent state dynamics. Nevertheless, I have some questions as follows:

1- The first motivation of the paper to define the VSDN is that the methods based on the neural ODEs cannot model complicated time-series. I wonder why is this claim true? When the underlying latent state is an ODE, the stochasticity of the time-series could be modeled via rich conditional observation distributions. Why do we need to have a double stochasticity, one in latent state and one in the observation data?

2- How many dimensions can VSDN handle? I am wondering how it will work in a high dimensional regime (say dim is several thousands). How many dimension are used for states and observation data?

3- As the VSDN is using SDEs for the latent states, I wonder if it can quantify the uncertainty of the inferred latent state trajectories?

4- Why in eq. (13), $h_{t,pre}$ and $h_t$ are summed? Could they be concatenated instead of summation?

5- Minor comment: I guess $W_t$ in eq. (2) refers to the Wiener process, right? But there is no definition on the text. So, the authors should define it in the paper so that the readers know what it is.


########## EDIT ##########
The authors have addressed all my questions.

---

> ### Author Response · Authors · 2020-11-23
> **Response to AnonReviewer2**
>
> **Q1: Why do we need to have a double stochasticity, one in latent state and one in the observation data?**
>
> ----  If we do not introduce the latent state, the stochastic process of the data can only be modeled by the parameterization of the conditional observation distribution. But in many cases, it is hard to find good parameterization of the distribution, especially when the data distribution is very complicated. Classic graphical models (i.e. Mixture Model) solve this problem by introducing a discrete-value latent variable.  Similarly, when we introduce the latent state into our model, we are able to define a more general and flexible parameterization of the data distribution (as the integral in Eq. (4)) under the same conditional observation distribution (In Neural ODEs, it is $P(y_i|y_{1:i-1})$, in VSDN and NeuralSDE, it is $P(y_i|y_{1:i-1}, X_{t_i}))$.
>
> **Q2: How many dimensions can VSDN handle?**
>
>  ----  In principle, our modal is quite scalable as a continuous-time stochastic RNN. In our motion experiments, the dimension of each data frame is 62 in MoCap dataset and 51 in h36m dataset. We also run an experiment on high-dimensional (above 1k) data. Our model has similar mean square errors with baselines but has much better NLL, which indicates that our model can better estimate the stochastic process of the data. In our future works, we will extend our work on more challenging very high-dimensional data (e.g. video).
>
> **Q3: quantify the uncertainty of the inferred latent state trajectories**
>
> ----  The quantification of the uncertainty can be achieved by analyzing the learned SDEs (e.g. compute the distribution of the latent state by numerically solving Fokker-Planck equation).
>
> **Q4: Why in eq. (13),  $\overrightarrow{h}_{t,pre}$ and $\overleftarrow{h}_t$ are summed? Could they be concatenated instead of summation?**
>
> ----  In our model, the inference model and the generative model share the SAME drift network, which is a feed-forward network. If we concatenate the forward and backward features, the input’s dimension is not consistent. We have to define a DIFFERENT posterior drift network. But the number of parameters will increase and we don’t see obvious improvement in the experiment.  Thus, we apply summation to use the SAME drift network.
>
> **Q5: Wiener process in Eq (2).**
>
> ---- Yes. $W_t$ is the Wiener process. We will add the definition in the revision (under Eq. (2) in Page 3).
> > where $H_{\mathcal{G}}$ and $R_{\mathcal{G}}$ are the drift and diffusion functions of the latent SDE. $W_t$ denotes the a Wiener process, which is also called standard Brownian motion.

---

### Official Review · AnonReviewer3 · 2020-10-29
**Review of Learning Continuous-Time Dynamics by Stochastic Differential Networks**

**Rating:** 4
**Confidence:** 3

**Review:**

This paper introduces a latent variable model for high dimensional stochastic time-series. The model is akin to a VAE with RNNs that incorporate time-series data. The authors introduce two variants of the model, one which only contains a feedforward RNN (filtering) and another that contains feedforward and feedback RNNs (smoothing).  The authors use two inference procedures for the model, one the standard VAE, and the other importance weighted IWAE.  The work is reasonably clearly presented and the experiments are multiple data sets are a nice addition.


I have two primary concerns -- 1) I do not see how the choice of the two inference procedures (which makes up a fair portion of the submission) is well motivated, and  2) I find the arguments for them achieving state-of-the-art performance unconvincing.

for 1)
The use of these two inference methods feels ad hoc. Why not just choose one? Using importance waiting and the standard ELBO does not seem to add to the paper. I see the primary contribution of the paper to be the model, not the inference procedure. If there is an important distinction between the inference methods, and the authors feel that is important to presenting their work, I would devote more space to explaining why there are two approaches --  the details of the objectives themselves could be moved to an appendix if needed.

Additionally, regarding the inference procedure, it looks like a beta-VAE objective is used (equation 6), but the authors do not say why the hyperparameter beta is needed, and do not cite the Beta-VAE paper (Higgins et al. 2017)

For 2)  I would like to see more comparing the performance of this model to a stochastic high dimensional. The authors discuss neural ODEs, but there is no comparison against them for performance in the experiment section. If the comparison cannot be made for some specific reason, the authors should explain why.

Another model which comes to mind that might warrant some discussion/comparison is lfads (Sussillo et al. 2016).

One other point: I feel similarly to the choice of two models (VSDN-F and VSDN-S) as I do about the IWAE and VAE. I think it would be better to hone in on a stronger take home point -- do one of the methods (say, VSDN-S) achieve better performance than existing methods. Showing this on more tasks or compared to more models would, in my mind, make this a stronger submission.

Some other concerns:

It is hard to get anything out of figure two. The human skeletons are very small and there are many of them. If the authors wish to show how the distinct positions for each of the methods differ from ground truth, they should remove some of the panels (again, not sure here why the different inference procedure warrants a different panel. I don't think the authors want to communicate how the different inference procedures make an important difference, do they?) An additional task or more clear plot showing accuracy of recovered latents, or accuracy of generated time-series data would be useful.


Typo in the final sentence of the first paragraph of the experiments section. You say "two" but compare to three things.

---

> ### Author Response · Authors · 2020-11-23
> **Response to AnonReviewer3**
>
> **Q1: important distinction between the inference methods and objectives, and the motivations.**
>
> ---- VAE vs IWAE: In principle, IWAE can provide tighter evidence lower bound than VAE. In our experiments, the models trained by IWAE generally have better log-likelihood than those trained by VAE.
>
> ----  Filtering vs Smoothing: The motivation of using both inference models has been explained in the paragraph before and after Eq (13). The scenarios of using these two models both exist. Smoothing is used when both history and future observations are given and we want to infer the latent states of some missing data. Filtering is used when only the history observations are given. In principle, the latent state in inference model has dependency on the whole data sequence. Therefore, the smoothing model is more accurate than filtering model, as the filtering model infers the latent state based up till current data points, while the smoothing model infers it by using the whole data obervations.In experiments, it is shown that VSDN-S has equal or better performance than VSDN-F. Besides, the filtering model does not have the backward RNN component (which is used to access the future information). Compared with VSDN-S, VSDN-F has promising performance with less parameters and smaller model complexity.
>
>
> **Q2: $\beta$-VAE objective is used (equation 6), but the authors do not say why the hyperparameter beta is needed, and do not cite the Beta-VAE paper (Higgins et al. 2017)**
>
>  ---- Thank you for pointing out the misleading part. Beta is a hyperparameter to weight the impact of KL term (as in $\beta$-VAE). However, during the experiment, we fix $\beta$ to 1. So we still use the original VAE objective to train our model. We will clarify it in the revision and also cite the $\beta$-VAE paper. The revision is given as follows:
>
> >Page 3:
> “
> β is a hyper-parameter to weight the effect of the KL terms.  In this paper, we fix β as1.0 and $\mathcal{L}_{VAE}$ is the original VAE objective (Kingma & Welling, 2014).  In β-VAE (Higginset al., 2017; Burgess et al., 2018), it is shown that a larger β can encourage the model to learn more efficient and disentangled representation from the data.
> ”
>
>
> **Q3: The authors discuss neural ODEs, but there is no comparison against them for performance in the experiment section. If the comparison cannot be made for some specific reason, the authors should explain why.**
>
> ---- In the SOTA papers we compared with (GRU-ODE, ODE-RNN and LatentSDE), they have showed that NeuralODE (called LatentODE in the neuralODE paper (Chen et al., 2018)) has worse performance. Our experiments (not shown in the paper) are also consistent with their results that NeuralODE has worse performance. That was why we didn’t include NeuralODE as a SOTA to compare with. We will add LatentODE in the revised version. Besides, GRU-ODE and ODE-RNN are two extended instances of NeuralODEs.
>
>
> **Q4: Another model which comes to mind that might warrant some discussion/comparison is lfads (Sussillo et al. 2016).**
>
> ---- Thank you for pointing out this missing reference. We will include the discussion of LFADS into our paper. To our knowledge, LFADS is a discrete-time sequential variational auto-encoder. Our work is a continuous-time stochastic variational model for sporadic data. That was why we didn’t include it in our previous version.
>
> According to your comment, we add a discussion in Page 3:
> >“
> The  VAE  objective  has  been  widely  used  for  discrete-time  stochastic  recurrent  modals,  such  as LFADS (Sussillo et al., 2016), VRNN (Chung et al., 2015) and SRNN (Fraccaro et al., 2016).  The major difference between these models and our work is that we incorporate a continuous-time latent state into our model while the latent state of the discrete-time models evolves only at distinct and separate time slots.
> ”
>
> **Q5: It is hard to get anything out of figure two.**
>
> ---- Thank you for your suggestion. We will remove some panels and make the skeletons larger in Figure 2 (Page 7). Besides, we have provided the video of the synthesized skeletons in the supplementary materials.

---

### Official Review · AnonReviewer1 · 2020-10-29
**Good paper yielding an interesting sequential SDE ELBO**

**Rating:** 7
**Confidence:** 3

**Review:**

The paper introduces Variational Stochastic Differential Networks to filter and smooth sporadically observed time series.

The authors adopt a Bayesian perceptive on the smoothing problem for time series living a latent space and irregularly observed.
In particular, the random evolution of the process in latent space is clearly accounted for in the paper by embedding an SDE into an RNN.

The authors derive a variational loss for their model and describe in fact multiple losses with different improvements such as importance sampling. In the end the authors do report issues caused by excess variance in training and settle for a convex combination of the two losses they propose.

The paper becomes quite interesting in its experimental part as the authors show the superiority of their method as compared to previous approaches on data sets concerned with Mocap, synthetic OU process data and meteorological data.

---

> ### Author Response · Authors · 2020-11-23
> **Response to AnonReviewer1**
>
> ----  Thanks for acknowledging our effort in this paper.

---

### Public Comment · ~David_Duvenaud2 · 2020-11-10
**Incorrect claims about latent SDEs**

This might be our fault for writing an unclear paper, but this submission makes several incorrect claims about our closely related method, which this paper calls latentSDE:

"LatentSDE (Li et al., 2020) includes the randomness in the latent state trajectory by replacing ODE with SDE. However, LatentSDE still encodes the time series into the initial states. This strategy is impractical and inefficient in real world applications, as it requires the initial latent states to disentangle the property of the long sequence."

This was true of the latent ODE model (Rubanova et al., 2019), but not of the latent SDE model.  In our latent SDE model, the recognition network outputs both an approximate posterior over the initial latent state, but also outputs the dynamics of the rest of the SDE, implicitly controlling the approximate posterior over the entire trajectory.  This can be done in a local manner too, it just depends on the architecture of the encoder.

"It also requires the neural ODE or SDE to predict target values at any time by using
only the initial states as the feature input."

This is incorrect.  Target values can be predicted from the hidden state at any time.

"Furthermore, LatentODE, ODE2VAE and LatentSDE are offline models, as they have to read the irregular time series before prediction and interpolation."

This is true, but presumably every prediction and interpolation method has to read the time series.  I think the authors meant that the encoder used during training of these models can't be directly used for online filtering, while e.g. an RNN can.  I think this is a fair criticism, but not the one that is written.

---

> ### Author Response · Authors · 2020-11-17
> **Response to "incorrect claims about laten SDEs"**
>
> Q1: the recognition network outputs both an approximate posterior over the initial latent state, but also outputs the dynamics of the rest of the SDE, implicitly controlling the approximate posterior over the entire trajectory.
>
>  ----  We appreciate your help to point out our misunderstanding of your study. We have carefully checked your paper as well as the public repository again, and updated the structure of latentSDE in our experiments. The results of latentSDE have significant improvement (e.g. the NLLs of latentSDE in Table 2 are improved from +57 to -64). The updated results will be included in our revision and the claim will also be revised in our update version. The updated submission will be uploaded in stage 2.
>
>
> Q2: This is incorrect. Target values can be predicted from the hidden state at any time.
> ---- We didn’t claim it correctly in the paper. The target values of latentODE can be predicted from the hidden state at any time, which is generated from the initial state that encodes the information of the whole sequence. This claim is not valid for latentSDE as the recognition network also controls the dynamics of the whole SDE.
>
> Q3:  I think the authors meant that the encoder used during training of these models can't be directly used for online filtering, while e.g. an RNN can.
> --- That’s exactly what we mean. Sorry that we don’t claim it clearly in the paper. In the revision, we will revise this sentence. Again, we are very thankful for your suggestions that are helpful to further improve this paper.

---

### Author Response · Authors · 2020-11-25
**Summarization of updates**

We appreciate the experts for providing constructive and helpful comments and suggestions. We have updated our submission following the suggestions by the reviewers. We hope that our responses can solve the concerns for reviewers and we summarize the updates as follows.

* Remove the incorrect claims about LatentSDE and update the experiment results of LatentSDE.
* Add definition of the notation $W_t$ under Eq. (2) in Page 3.
* Add the discussion and citations of beta-VAE.
* Add experiment results of LatentODE in Section 4.
* Add discussion of discrete-time sequential VAE (e.g LFADS (Sussillo et al. 2016))
* Revise and make the skeletons larger in Figure 2.
* Add an example with detailed deduction and discussion in Appendix B to explain why we don’t include the latent state as the input of the diffusion function.
* Add a discussion in Section 3 to clearly describe the functionality of each component.
* Claim the motivation of filtering inference models more clearly.
* Add supplementary experiments on high-dimensional sequence (Appendix C.3).

---

### Decision · Program_Chairs · 2021-01-07
**Final Decision**

**Decision:**

Reject

**Comment:**

This paper sits right at the borderline: the reviewers agree that it is interesting and addresses a relevant problem. On the negative side, the presentation could be improved (including some incorrect claims), and the experiments could be strengthened (both in terms of baselines and datasets used). Ultimately, the paper will probably require another round of reviews before it is ready for publication.